# Faithful Embeddings for Knowledge Base Queries

**Haitian Sun**      **Andrew O. Arnold**[*]      **Tania Bedrax-Weiss**
**Fernando Pereira**      **William W. Cohen**
Google Research
{haitiansun,tbedrax,pereira,wcohen}@google.com
AWS AI
anarnld@amazon.com

## Abstract

The deductive closure of an ideal knowledge base (KB) contains exactly the logical queries that the KB can answer. However, in practice KBs are both incomplete and over-specified, failing to answer some queries that have real-world answers. *Query embedding* (QE) techniques have been recently proposed where KB entities and KB queries are represented jointly in an embedding space, supporting relaxation and generalization in KB inference. However, experiments in this paper show that QE systems may disagree with deductive reasoning on answers that do not require generalization or relaxation. We address this problem with a novel QE method that is more faithful to deductive reasoning, and show that this leads to better performance on complex queries to incomplete KBs. Finally we show that inserting this new QE module into a neural question-answering system leads to substantial improvements over the state-of-the-art. [2]

## 1 Introduction

The deductive closure of an ideal knowledge base (KB) contains exactly the logical queries that the KB can answer. However, in practice KBs are both incomplete and over-specified, failing to answer queries that have actual real-world answers. *Query embedding* (QE) methods extend logical queries to incomplete KBs by representing KB entities and KB queries in a joint embedding space, supporting relaxation and generalization in KB inference [9, 10, 25, 17]. For instance, graph query embedding (GQE) [10] encodes a query $q$ and entities $x$ as vectors such that cosine distance represents $x$'s score as a possible answer to $q$. In QE, the embedding for a query $q$ is typically built compositionally; in particular, the embedding for $q = q_1 \wedge q_2$ is computed from the embeddings for $q_1$ and $q_2$. In past work, QE has been useful for answering overconstrained logical queries [25] and querying incomplete KBs [9, 10, 17].

Figure 1 summarizes the relationship between traditional KB embedding (KBE), query embedding (QE), and logical inference. Traditional logical inference enables a system to find deductively *entailed* answers to queries; KBE approaches allow a system to *generalize* from explicitly-stored KB tuples to similar tuples; and QE methods combine both of these ideas, providing a soft form of logical entailment that generalizes.

We say that a QE system is *logically faithful* if it behaves similarly to a traditional logical inference system with respect to entailed answers. In this paper, we present experiments illustrating that QE systems are often *not* faithful: in particular, experiments with the state-of-the-art QE system Query2Box [17] show that it performs quite poorly in finding logically-entailed answers. We conjecture this is because models that generalize well do not have the capacity to model all the

---

[*]Work done while at Google Research.

[2]Code available at `https://github.com/google-research/language`

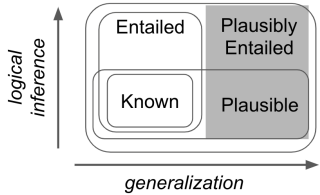

KB embedding (KBE) methods generalize from *known* KG facts to *plausible* ones, and logical inference computes answers to compositional queries that are *entailed* by known facts. Query embedding (QE) combines both of these tools for extending a set of known facts, by finding answers to a query that are *plausibly entailed* by known facts.

Figure 1: Overview of differences between KBE and QE. Shaded area indicates the kinds of test cases used in prior studies of QE.

information in a large KB accurately, unless embeddings are impractically large. We thus propose two novel methods for improving faithfulness while preserving the ability to generalize. First, we implement some logical operations using neural retrieval over a KB of embedded triples, rather than with geometric operations in embedding space, thus adding a non-parametric component to QE. Second, we employ a randomized data structure called a count-min sketch to propagate scores of logically-entailed answers. We show that this combination leads to a QE method, called EmQL (Embedding Query Language) which is differentiable, compact, scalable, and (with high probability) faithful. Furthermore, strategically removing the sketch in parts of the QE system allows it to generalize very effectively.

We show that EmQL performs dramatically better than Query2Box on logically-entailed answers, and also improves substantially on complex queries involving generalization. Finally we show that inserting EmQL into a natural language KB question-answering (KBQA) system leads to substantial improvements over the experimental state-of-the-art for two widely-used benchmarks, MetaQA [29] and WebQuestionsSP [27].

The main contributions of this work are: (1) a new QE scheme with expressive set and relational operators, including those from previous QE schemes (set intersection, union, and relation following) plus a "relational filtering" operation; (2) a new analysis of QE methods showing that previous methods are not faithful, failing to find entities logically entailed as answers; (3) the first application of QE as a module in a KBQA system; and (4) evidence that this module leads to substantial gains over the prior state-of-the-art on two widely-used benchmarks, thanks to its superior faithfulness.

## 2 Related work

**KBE and reasoning.** There are many KB embedding (KBE) methods, surveyed in [26]. Typically KBE methods generalize a KB by learning a model that scores the plausibility of a potential KB triple $r(x, y)$, where $r$ is a KB relation, $x$ is a head (aka subject) entity, and $y$ is a tail (aka object) entity. In nearly all KBE models, the triple-scoring model assumes that every entity $x$ is represented by a vector $\mathbf{v}_x$.

Traditional query languages for symbolic KBs do support testing whether a triple is present in a KB, but also allow expressive compositional queries, often queries that return sets of entities. Several previous works also propose representing *sets* of entities with embeddings [23, 24, 28]; box embeddings [24], for instance, represent sets with axis-parallel hyperrectangles.

Many KBE models also support *relation projection*, sometimes also called *relation following*. *Relation following* [3] maps a set of entities $X$ and a set of relations $R$ to a set of entities related to something in $X$ via some relation in $R$: here we use the notation $X.follow(R) \equiv \{y \mid \exists x \in X, r \in R : r(x, y)\}$. Many KBEs naturally allow computation of some soft version of relation following, perhaps restricted to singleton sets.[3] However, most KBE methods give poor results when relation following operations are composed [9], as in computing $X.follow(R_1).follow(R_2)$. To address this, some KBE systems explicitly learn to follow a path (aka chain) of relations [9, 11, 6].

Extending this idea, the graph-query embedding (GQE) method [10] defined a query language containing both relation following and set intersection. In GQE inputs and output to the relation

following operation are entity sets, defined by cosine-distance proximity to a central vector. More recently, the Query2Box [17] method varied GQE by adopting a box embedding for sets, and also extended GQE by including a set union operator. In Query2Box unions are implemented by rewriting queries into a normal form where unions are used only as the outermost operation, and then representing set unions as unions of the associated boxes.

Quantum logic [21] is another neural representation scheme, which might be considered a query language. It does not include relation following, but is closed under intersection and negation, and approximately closed under union (via computation of an upper bound on the union set.)

Other studies [7, 16] use logical constraints such as transitivity or implication to improve embeddings. Here we go in the opposite direction, from KBE to reasoning, answering compositional queries in an embedded KB that is formed in the absence of prior knowledge about relations.

**Sparse-matrix neural reasoning.** An alternative to representing entity sets with embeddings is to represent sets with "$k$-hot" vectors. Set operations are easily performed on $k$-hot vectors[4] and relation following can be implemented as matrix multiplication [2]. Such "localist" representations can exactly emulate logical, hence faithful, reasoning systems. However, they do not offer a direct way to generalize because entities are just (represented by) array indices.

## 3 Faithful queries on an embedded KB

**Background and notation.** The query language EmQL operates on weighted sets of entities. Let $U$ be the set of all entities in a KB. A weighted set $X \subseteq U$ is *canonically encoded* as a $k$-hot vector $\mathbf{v}_X \in \mathbb{R}^N$, where $N = |U|$ and $\mathbf{v}_X[i]$ holds the non-negative real *weight* of element $i$ in $X$. However the $k$-hot encoding is very inefficient if $N$ is large, which we address later. EmQL relies on a learned embedding $\mathbf{e}_i \in \mathbb{R}^d$ for each entity $i$, which together form the matrix $\mathbf{E} \in \mathbb{R}^{d \times N}$ of entity embeddings. A weighted set $X$ will be represented by a pair consisting of (1) a dense vector derived from its entity embeddings $\{\mathbf{e}_i\}$, $i \in X$, plus an efficient sparse representation of the weights $\mathbf{v}_X[i]$.

In addition to (weighted) set intersection, union, and difference, which are common to many KBE models, EmQL implements two operators for relational reasoning: *relation following* and *relational filtering*. EmQL also supports a limited form of set difference (see Supplemental Material C.) In this section, we will start by discussing how to encode and decode sets with EmQL representations, and then discuss the operators in EmQL for relational reasoning.

**Representing sets.** We would like to represent entity sets with a scheme that supports generalization, but also allows for precisely encoding weights of sets that are defined by compositional logic-like operations. Our representation will assume that sets are of limited cardinality, and contain "similar" entities (as defined below).

We represent a set $X$ with the pair $(\mathbf{a}_X, \mathbf{b}_X)$, $\mathbf{a}_X = \sum_i \mathbf{v}_X[i] \, \mathbf{e}_i$, $\mathbf{b}_X = \mathbf{S}_H(\mathbf{v}_X)$ where $\mathbf{a}_X$ is the weighted centroid of elements of X that identifies the general region containing elements of $X$, and $\mathbf{b}_X$ is an optional count-min sketch [4], which encodes additional information on the weights of elements of $X$. Count-min sketches [4] are a widely used randomized data structure that can approximate the vector $\mathbf{v}_X$ with limited storage. Supplemental Material B summarizes more technical details, but we summarize count-min sketches below. Our analysis largely follows [5].

Let $h$ be a hash function mapping $\{1, \ldots, N\}$ to a smaller range of integers $\{1, \ldots, N_W\}$, where $N_W \ll N$. The *primitive sketch of $\mathbf{v}_X$ under $h$*, written $\mathbf{s}_h(\mathbf{v}_X)$, is a vector such that

$$\mathbf{s}_h(\mathbf{v}_X)[j] = \sum_{i:h(i)=j} \mathbf{v}_X[i]$$

Algorithmically, this vector could be formed by starting with an all-zero's vector of length $N_W$, then looping over every pair $(i, w_i)$ where $w_i = \mathbf{v}_X[i]$ and incrementing each $\mathbf{s}_h[j]$ by $w_i$.

A primitive sketch $\mathbf{s}_h$ contains some information about $\mathbf{v}_X$: to look up the value $\mathbf{v}_X[i]$, we could look up $\mathbf{s}_h[h(i)]$, and this will have the correct value if no other set element $i'$ hashed to the same location. We can improve this by using multiple hash functions. Specifically, let $H = \{h_1, \ldots, h_{N_D}\}$ be a list of $N_D$ hash functions mapping $\{1, \ldots, N\}$ to the smaller range of integers $\{1, \ldots, N_W\}$. The

*count-min sketch $\mathbf{S}_H(\mathbf{v}_X)$ for a $\mathbf{v}_X$ under $H$* is a matrix such that each row $j$ is the primitive sketch of $\mathbf{v}_X$ under $h_j$. This sketch is an $N_W \times N_D$ matrix, where $N_W$ is called the sketch *width* and $N_D$ is called the sketch *depth*.

Let $\mathbf{b}_X = \mathbf{S}_H(\mathbf{v}_X)$ be the count-min sketch for $X$. To "look up" (approximately recover) the value of $\mathbf{v}_X[i]$, we compute the quantity

$$CM(i, \mathbf{b}_X) \equiv \min_{j=1}^{N_D} \mathbf{b}_X[j, h_j(i)]$$

In other words, we look up the hashed value associated with $i$ in each of the $N_D$ primitive sketches, and take the minimum value. The "look up" of the count-min sketch provides the following probabilistic guarantee, as proved in Supplementary Material B.

**Theorem 1** *Let $\mathbf{b}_X$ be a count-min sketch for $X$ of depth $N_D$ and with $N_W > 2|X|$, and let $C \supseteq X$. If $N_D > \log_2 \frac{|C|}{\delta}$ then with probability at least 1-$\delta$, $X$ can be recovered from $\mathbf{b}_X$ using $C$.*

To reconstruct a set from this encoding, we first take the $k$ elements with highest dot product $\mathbf{a}_X^T \mathbf{e}_i$, where $k$ is a fixed hyperparameter. This is done efficiently with a maximum inner product search [15] (MIPS), which we write $\text{TOP}_k(\mathbf{a}_X, \mathbf{E})$.[5] These top $k$ elements are then filtered by the count-min sketch, resulting in a sparse (no more than $k$ non-zeros) *decoding* of the set representation

$$\hat{\mathbf{v}}_X[i] = \begin{cases} CM(i, \mathbf{b}_X) \cdot \text{softmax}(\mathbf{a}_X^T \mathbf{e}_i) & \text{if } i \in \text{TOP}_k(\mathbf{a}_X, \mathbf{E}) \\ 0 & \text{else} \end{cases}$$

The two pairs of the centroid-sketch representation are complementary. The region around a centroid will usually contain entities with many similar properties, for example "US mid-size cities," or "Ph.D. students in NLP": conceptually, it can be viewed as defining a *soft type* for the entities in $X$. However, simple geometric representations like centroids are not expressive enough to encode arbitrary sets $X$, like "Ph.D. students presenting papers in session $z$". Count-min sketches do allow arbitrary weights to be stored, but may return incorrect values (with low probability) when queries. However, in this scheme the sketch is only queried for $k$ candidates close to the centroid, so it is possible to obtain very low error probabilities with small sketches (discussed later).

The centroid-sketch representation does assume that all elements of the same set are similar in the sense that they all have the same "soft type"—i.e., are all in a sphere around a specific centroid. It also assumes that sets are of size no more than $k$. (Note the radius of the sphere is not learned—instead $k$ is simply a hyperparameter.)

**Faithfulness.** Below we define compositional operations (like union, intersection, etc) on centroid-sketch set representations. A representation produced this way is associated with a particular logical definition of a set $X$ (e.g., $X = Z_1 \cup Z_2$), and we say that the representation is *faithful* to that definition to the extent that it yields the appropriate elements when decoded (which can be measured experimentally).

Experimentally sketches improve the faithfulness of EmQL. However, *the sketch part of a set representation is optional*—specifically it can be replaced with a vacuous sketch that returns a weight of 1.0 whenever it is queried.[6] Removing sketches is useful when one is focusing on generalization.

**Intersection and union**. Set interesection and union of sets $A$ and $B$ will be denoted as $(\mathbf{a}_{A \cap B}, \mathbf{b}_{A \cap B})$ and $(\mathbf{a}_{A \cup B}, \mathbf{b}_{A \cup B})$, respectively. Both operations assume that the soft types of $A$ and $B$ are similar, so we can define the new centroids as

$$\mathbf{a}_{A \cap B} = \mathbf{a}_{A \cup B} = \frac{1}{2}(\mathbf{a}_A + \mathbf{a}_B)$$

To combine the sketches, we exploit the property (see Supplemental Materials) that if $\mathbf{b}_A$ and $\mathbf{b}_B$ are sketches for $A$ and $B$ respectively, then a sketch for $A \cup B$ is $\mathbf{b}_A + \mathbf{b}_B$, and the sketch for $A \cap B$ is $\mathbf{b}_A \odot \mathbf{b}_B$ (where $\odot$ is Hadamard product). Hence

$$\mathbf{b}_{A \cap B} = \mathbf{b}_A \odot \mathbf{b}_B \quad \mathbf{b}_{A \cup B} = \mathbf{b}_A + \mathbf{b}_B$$

**Relational following.** As noted above, relation following takes a set of entities $X$ and a set of relations $R$ and computes the set of entities related to something in $X$ via some relation in $R$:

$$X.follow(R) \equiv \{y \mid \exists r \in R, x \in X : r(x, y)\}$$

where "$r(x, y)$" indicates that this triple is in the KB (other notation is listed in Supplemental Material A.) For example, to look up the headquarters of the Apple company one might compute $Y = X.follow(R)$ where $X$ and $R$ are singleton sets containing "*Apple_Inc*" and "*headquarters_of*" respectively, and result set $Y = \{Cupertino\}$.

Relation following is implemented using an embedding matrix $\mathbf{K}$ for KB triples that parallels the element embedding matrix $\mathbf{E}$: for every triple $t = r(x, y)$ in the KB, $\mathbf{K}$ contains a row $\mathbf{r}_t = [\mathbf{e}_r; \mathbf{e}_x; \mathbf{e}_y]$ concatenating the embeddings for $r$, $x$, and $y$. To compute $Y = X.follow(R)$ first we create a query $\mathbf{q}_{R,X} = [\lambda \cdot \mathbf{a}_R; \mathbf{a}_X; \mathbf{0}]$ by concatenating the centroids for $R$ and $X$ and padding it to the same dimension as the triple embeddings (and $\lambda$ is a hyper-parameter scaling the weight of the relation). Next using the query $\mathbf{q}_{R,X}$, we perform a MIPS search against all triples in KB $\mathbf{K}$ to get the top $k$ triples matching this query, and these triples are re-scored with the sketches of $X$ and $R$. Let $\mathbf{r}_t = [\mathbf{e}_{r_i}; \mathbf{e}_{x_j}; \mathbf{e}_{y_\ell}]$ be the representation of retrieved triple $t = r_i(x_j, y_\ell)$. Its score is

$$s(\mathbf{r}_t) = \text{CM}(i, \mathbf{b}_R) \cdot \text{CM}(j, \mathbf{b}_X) \cdot \text{softmax}(\mathbf{q}_{R,X}^T \mathbf{r}_t)$$

We can then project out the objects from the top $k$ triples as a sparse $k$-hot vector:

$$\hat{\mathbf{v}}_Y(\ell) = \sum_{\mathbf{r}_t \in \text{TOP}_k(\mathbf{q}_{R,X}, \mathbf{K}), t=\_(\_, y_\ell)} s(\mathbf{r}_t)$$

Finally $\hat{\mathbf{v}}_Y$ is converted to a set representation $(\mathbf{a}_Y, \mathbf{b}_Y)$, which represents the output of the operation, $Y = X.follow(R)$. The triple store used for implementing *follow* is thus a kind of key-value memory network [14], augmented with a sparse-dense filter in the form of a count-min sketch.

**Relational filtering.** Relational filtering, similar to an existential restriction in description logics, removes from $X$ those entities that are not related to something in set $Y$ via some relation in $R$:

$$X.filter(R, Y) \equiv \{x \in X \mid \exists r \in R, y \in Y : r(x, y)\}$$

For example, $X.filter(R, Y)$ would filter out the companies in $X$ whose headquarters are not in Cupertino, if $R$ and $Y$ are as in the previous example. Relational filtering is implemented similarly to *follow*. For $X.filter(R, Y)$, the query must also be aware of the objects of the triples, since they should be in the set $Y$. The query vector is thus $\mathbf{q}_{R,X,Y} = [\lambda \cdot \mathbf{a}_R; \mathbf{a}_X; \mathbf{a}_Y]$. Again, we perform a retrieval using query $\mathbf{q}_{R,X,Y}$, but we filter with subject, relation, and object sketches $\mathbf{b}_R, \mathbf{b}_X, \mathbf{b}_Y$, so the score of an encoded triple $\mathbf{r}_t$ is

$$s(\mathbf{r}_t) = \text{CM}(i, \mathbf{b}_R) \cdot \text{CM}(j, \mathbf{b}_X) \cdot \text{CM}(\ell, \mathbf{b}_Y) \cdot \text{softmax}(\mathbf{q}_{R,X,Y}^T \mathbf{r}_t)$$

The same aggregation strategy is used as for the *follow* operation, except that scores are aggregated over the subject entities instead of objects.

**Unions in EmQL vs Query2Box.** By construction, all EmQL operations are closed under composition, because they all take and return the same sparse-dense representations, and the computation graph constructed from an EmQL expression is similar in size and structure to the original EmQL expression. We note this differs from Query2Box, where union is implemented by rewriting a query into a normal form. A disadvantage of the Query2Box normal-form approach is that the normal form can be exponentially larger than the original expression.

However, a disadvantage of EmQL's approach is that unions are only allowed between sets of similar "soft types". In fact, EmQL's centroid-sketch representation will not compactly encode *any* set of sufficiently diverse entities: in a small embedding space, a diverse set like $X = \{kangaroo, ashtray\}$ will have a centroid far from any element, so a top-$k$ MIPS query with small $k$ would have low recall. This limitation of EmQL could be addressed by introducing a second normal-form disjunction operation that outputs a union of centroid-sketch representations, much as Query2Box's disjunction outputs a union of boxes—however, we leave such an extension as a topic for future work.

**Size and density of sketches.** Although the centroid-based geometric constraints are not especially expressive, we note that EmQL's sparse-dense representation can still express sets accurately, as long as the $k$-nearest neighbor retrieval has good recall. Concretely, consider a set $A$ with $|A| = m$ and

sparse-dense representation $(\mathbf{a}_A, \mathbf{b}_A)$. Suppose that $k = cm$ ensures that all $m$ elements of $A$ are retrieved as $k$-nearest neighbors of $\mathbf{a}_A$; in other words, retrieval precision may be as low as $1/c$. By Theorem 2 in the Supplementary Materials, a sketch of size $2m \log_2 \frac{cm}{\delta}$ will recover *all* the weights in $A$ with probability at least $1 - \delta$.

In our experiments we assume sets are of size $m < 100$, and that $c = 10$. Using 32 numbers per potential set member leads to $\delta \approx \frac{1}{50}$ and a sketch size of about 4k. Put another way, sets of 100 elements require about as much storage as the BERT [8] contextual encoding of 4 tokens; alternatively the sketch for 100 elements requires about 1/4 the storage of 100 embeddings with $d = 128$.[7]

It is also easy to see that for a set of size $m$, close to half of the numbers in the sketch will have non-zero values. Thus only a moderate savings in space is obtained by using a sparse-matrix data structure: it is quite practical to encode sketches with GPU-friendly dense-tensor data structures.

**Loss function.** This representation requires entities that will be grouped into sets to be close in embedding space, so entity embeddings must be trained to have this property—ideally, for all sets that arise in the course of evaluating queries. In the training process we use to encourage this property, an example is a query (such as "$\{Apple\_Inc\}.follow(\{headquarters\_of\} \cup \{Sunnyvale\}$") and a target output set $Y$. Evaluation of the query produces an approximation $\hat{Y}$, encoded as $(\hat{\mathbf{a}}_Y, \hat{\mathbf{b}}_Y)$, and the goal of training is make $\hat{Y}$ approximate $Y$.

Let $\mathbf{v}_Y$ be the canonical $k$-hot encoding of $Y$. While the sketches prevent an element $y' \notin \hat{Y}$ from getting too high a score, the top-$k$ operator used to retrieve candidates only has high recall if the elements in $\hat{Y}$ are close in the inner product space. We thus train embeddings to minimize

$$\text{cross\_entropy}(\text{softmax}(\hat{\mathbf{a}_Y}^T, \mathbf{E}), \mathbf{v}_Y / \|\mathbf{v}_Y\|_1)$$

Note that this objective ignores the sketch[8], so it forces the dense representation to do the best job possible on its own. In training $\hat{Y}$ can be primitive set, or the result of a computation (see § 4.1).

# 4 Experiments

We evaluate EmQL first intrinsically for its ability to model set expressions [10], and then extrinsically as the reasoning component in two multi-hop KB question answering benchmarks (KBQA).

## 4.1 Learning to reason with a KB

**Generalization.** To evaluate performance in generalizing to plausible answers, we follow the procedure of Ren et al. [17] who considered nine different types of queries, as summarized in Table 1, and data automatically constructed from three widely used KB completion (KBC) benchmarks. Briefly, to evaluate performance for QE, Ren et al. first hold out some triples from the KB for validation and test, and take the remaining triples as the *training KB*. Queries are generated randomly using the query templates of Table 1. The gold answer for a query is the traditional logical evaluation on the *full KB*, but the QE system is trained to approximate the gold answer using only the smaller *training KB*. Queries used to evaluate the system are also constrained to *not* be fully answerable using only logical entailment over the training KB. For details, see Supplementary Materials D.

|  | Query Template |  | Query Template |
|---|---|---|---|
| 1p | $X.follow(R)$ | ip | $(X_1.follow(R_1) \cap X_2.follow(R_2)).follow(R)$ |
| 2p | $X.follow(R_1).follow(R_2)$ | pi | $X_1.follow(R_1).follow(R_2) \cap X_2.follow(R_3)$ |
| 3p | $X.follow(R_1).follow(R_2).follow(R_3)$ | 2u | $X_1.follow(R_1) \cup X_2.follow(R_2)$ |
| 2i | $X_1.follow(R_1) \cap X_2.follow(R_2)$ | up | $(X_1.follow(R_1) \cup X_2.follow(R_2)).follow(R)$ |
| 3i | $X_1.follow(R_1) \cap X_2.follow(R_2) \cap X_3.follow(R_3)$ |  |  |

Table 1: Nine query templates used. Query2Box is trained on templates 1p, 2p, 3p, 2i, and 3i. EmQL is trained on a variation of 1p and set intersection.

| | 1p | 2p | 3p | 2i | 3i | ip | pi | 2u | up | Avg | FB15k Avg | NELL Avg |
|---|---|---|---|---|---|---|---|---|---|---|---|---|
| Generalization on FB15k-237 | | | | | | | | | | | FB15k | NELL |
| GQE | 40.5 | 21.3 | 15.5 | 29.8 | 41.1 | 8.5 | 18.2 | 16.9 | 16.3 | 23.1 | 38.7 | 24.8 |
| Q2B | **46.7** | 24 | 18.6 | 32.4 | 45.3 | 10.8 | 20.5 | **23.9** | 19.3 | 26.8 | 48.4 | 30.6 |
| $+d$=2000 | 37.2 | 20.7 | 19.4 | 22.6 | 37.1 | 9.7 | 16.8 | 20.0 | 17.8 | 22.4 | 34.5 | 23.4 |
| EmQL (ours) | 37.7 | **34.9** | **34.3** | **44.3** | **49.4** | **40.8** | **42.3** | 8.7 | 28.2 | **35.8** | **49.5** | **46.8** |
| − sketch | 43.1 | 34.6 | 33.7 | 41.0 | 45.5 | 36.7 | 37.2 | 15.3 | **32.5** | 35.5 | 48.6 | **46.8** |
| Entailment on FB15k-237 | | | | | | | | | | | FB15k | NELL |
| Q2B | 58.5 | 34.3 | 28.1 | 44.7 | 62.1 | 11.7 | 23.9 | 40.5 | 22.0 | 36.2 | 43.7 | 51.1 |
| $+d$=2000 | 50.7 | 30.1 | 26.1 | 34.8 | 55.2 | 11.4 | 20.6 | 32.8 | 21.5 | 31.5 | 38.3 | 43.7 |
| EmQL (ours) | **100.0** | **99.5** | **94.7** | **92.2** | **88.8** | **91.5** | **93.0** | **94.7** | **93.7** | **94.2** | **91.4** | **98.8** |
| − sketch | 89.3 | 55.7 | 39.9 | 62.9 | 63.9 | 51.9 | 54.7 | 53.8 | 44.7 | 57.4 | 55.5 | 82.5 |

Table 2: Hits@3 results on the Query2Box datasets. Please see Supplementary Materials for full results on FB15k and NELL995 datasets and for mean reciprocal rank results.

Query2Box is trained on examples from only five reasoning tasks (1p, 2p, 3p, 2i, 3i), with the remainder held out to measure the ability to generalize to new query templates.[9] EmQL was trained on only two tasks: *relational following* (a variation of 1p), and *set intersection*. Specifically we define a "basic set" $X$ to be any set of entities that share the same property $y$ with relation $r$, i.e. $X = \{x|r(x,y)\}$. In training EmQL to answer intersection queries ($X_1 \cap X_2$), we let $X_1$ and $X_2$ be non-disjoint basic sets, and for relational following (1p), $X$ is a basic set and $R$ a singleton relation set. Training, using the method proposed in §3, produces entity and relation embeddings, and queries are then executed by computing the EmQL representations for each subexpression in turn.[10] Since we are testing generalization, rather then entailment, we replace $\mathbf{b}_{\hat{Y}}$ with a vacuous all-ones count-min sketch in the final set representation for a query (but not intermediate ones).

We compare EmQL to two baseline models: GQE [10] and Query2Box (Q2B) [17]. The numbers are shown in Table 2. Following the Query2Box paper [17] we use $d = 400$ for their model and report Hits@3 (see Supplementary Materials D for other metrics). For EmQL, we use $d = 64$, $k = 1000$, $N_W = 2000$ and $N_D = 20$ throughout. In this setting, our model is slightly worse than Query2Box for the 1p queries, much worse for the 2u queries, and consistently better on all the more complex queries. EmQL's difficulties with the 2u queries are because of its different approach to implementing union, in particular the kangaroo-ashtray problem discussed in §3.

**Entailment.** To test the ability to infer logically entailed answers, EmQL and Q2B were trained with the full KB instead of the training KB, so only reasoning (not generalization) is required to find answers. As we argue above, it is important for a query language to be also be faithful to the KB when answering compositional logical queries. The results in Table 2 show EmQL dramatically outperforms Q2B on all tasks in this setting, with average Hits@3 raised from 36-51 to the 90's.

To see if larger embeddings would improve Q2B's performance on entailment tasks, we increased the dimension size to $d = 2000$, and observed a decrease in performance, relative to the tuned value $d = 400$ [17].[11] In the ablation experiment(EmQL−sketch), we remove the sketch and only use the centroid to make predictions. The results are comparable for generalization, but worse for entailment.

## 4.2 Question answering

To evaluate QE as a neural component in a larger system, we followed [3] and embed EmQL as a reasoning component in a KBQA model. The reasoner of [3], ReifKB, is a sparse-matrix "reified KB" rather than a QE method, which does not generalize, but is perfectly faithful for entailment questions. In this section we evaluate replacing ReifKB with EmQL.

ReifKB was evaluated on two KBQA datasets, MetaQA [29] and WebQuestionsSP [27], which access different KBs. For both datasets, the input to the KBQA system is a question $q$ in natural language and a set of entities $X_q$ mentioned in the question, and the output is a set of answers $Y$

.

(but no information is given about the latent logical query that produces $Y$.) EmQL's set operations were pre-trained for each KB as in § 4.1, and then the KB embeddings were fixed while training the remaining parts of the QA model. Please see the Supplementary Materials for details.

**The MetaQA model**. The MetaQA datasets [29] contain multi-hop questions in the movie domain that are answerable using the WikiMovies KB [13], e.g., "When were the movies directed by Christopher Nolan released?"). One dataset (here called MetaQA2) has 300k 2-hop questions and the other (MetaQA3) has 300k 3-hop questions. We used similar models as those used for ReifKB. The model[12] for 2-hop questions is given on the left of Table 3, where $W_1$ and $W_2$ are learned parameters, $\mathbf{b}_{\mathbb{I}}$ is a vacuous sketch, and $encode(q)$ is obtained by pooling the (non-contextual) embeddings of words in $q$. The 3-hop case is analogous.[13]

| MetaQA | WebQuestionsSP |
|---|---|
| $\hat{Y} = X_q.follow(R_1).follow(R_2) - X_q$ | $X_1 = X_q.follow(R_1^e)$ |
| $R_1 = (\mathbf{a}_1, \mathbf{b}_{\mathbb{I}}),\ \mathbf{a}_1 = W_1^T encode(q)$ | $X_2 = X_q.follow(R_1^{cvt}).follow(R_2^e)$ |
| $R_2 = (\mathbf{a}_2, \mathbf{b}_{\mathbb{I}}),\ \mathbf{a}_2 = W_2^T encode(q)$ | $\hat{Y} = X_1 \cup X_2 \cup (X_1 \cup X_2).filter(R_3, Z)$ |

Table 3: EmQL models for MetaQA and WebQuestionsSP datasets.

**The WebQuestionsSP model**. This dataset [27] contains 4,737 natural language questions generated from Freebase. Questions in WebQuestionsSP are a mixture of 1-hop and 2-hop questions, sometimes followed by a relational filtering operation, which are answerable using a subset of FreeBase. The intermediate entities of 2-hop questions are always "compound value type" (CVT) entities—entities that do not have names, but describe $n$-ary relationships between entities. For example, the question "Who is Barack Obama's wife?" might be answered with the query $X_q.follow(R_1).follow(R_2).filter(R_3,Z)$, where $X_q = \{Barack\_Obama\}$, $R_1$, $R_2$, and $R_3$ are the relations *has_marriage*, *spouse*, and *gender*, and $Z$ is the set $\{female\}$. Here $X_q.follow(R_1)$ produces a CVT node representing a marriage relationship for a couple. The model we use (see Table 3, right) is similar to MetaQA model, except that the final stage is a union of several submodels—namely, chains of one and two follow operations, with or without relational filtering. The submodels for the $R$'s also similar to those for MetaQA, except that we used a BERT [8] encoding of the question, and augmented the entity embeddings with pre-trained BERT representations of their surface forms (see Supplementary Materials.)

The model in ReifKB [3] does not support relational filtering[14], so for it we report results with the simpler model $\hat{Y} = X_1 \cup X_2$. Of course the EmQL models also differ in being QE models, so they model relations as centroids in embedding space instead of $k$-hot vectors.

**Experimental results.** In addition to ReifKB [3], we report results for GRAPH-Net [19] and PullNet [20], which are Graph-CNN based methods. EmbedKGQA [18] is the current state-of-the-art model that applies KB embeddings ComplEx [22] in KBQA. Since our models make heavy use of the *follow* operation, which is related to a key-value memory, we also compare to a key-value memory network baseline [13]. The results are shown on Table 4.

On MetaQA3 and WebQSP datasets we exceed the previous state-of-the-art by a large margin (7.7% and 5.8% hits@1 absolute improvement), and results on MetaQA2 are comparable to the previous state-of-the-art. We also consider two ablated versions of our model, EmQL-sketch and EmQL-filter. EmQL-filter uses the same model as used in ReifKB [3], but still improves over ReifKB significantly, showing the value of coupling learning with a QE system rather than a localist KB. EmQL-sketch disables sketches throughout (rather than only in the submodels for the $R$'s), and works consistently worse than the full model for all datasets. Thus it underscores the value of *faithful QE* for KBQA tasks. (Notice that errors in computing entailed answers will appear to the KBQA system as noise in the end-to-end training process.) Finally, Figure 2 (right) shows that performance of EmQL-sketch is improved only slightly with much larger embeddings, also underscoring the value of the sketches.

As noted above, the QE component was first pre-trained on synthetic queries (as in § 4.1), and then the QA models were trained with fixed entity embeddings. We also jointly trained the KB embeddings

|           | MetaQA2 | MetaQA3 | WebQSP |
|-----------|---------|---------|--------|
| KV-Mem    | 82.7    | 48.9    | 46.7   |
| GRAFT-Net | 94.8    | 77.7    | 70.3   |
| PullNet   | **99.9**| 91.4    | 69.7   |
| EmbedKGQA | 98.8    | 94.8    | 66.6   |
| ReifKB    | 81.1    | 72.3    | 52.7   |
| EmQL (ours) | 98.6  | **99.1**| **75.5** |
| − filter  | –       | –       | 65.2   |
| − sketch  | 70.3    | 60.9    | 53.2   |

Table 4: Hits@1 of WebQuestionsSP, MetaQA2, and MetaQA3. GRAFT-Net and PullNet were re-run on WebQuestionsSP with oracle sets of question entities $X_q$.

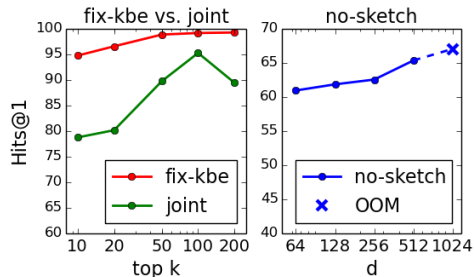

Figure 2: Hits@1 on MetaQA3. **Left:** Jointly training (joint) or fixed QE (fix-kbe) varying $k$ for top $k$ retrieval. **Right:** Varying $d$ for EmQL-sketch.

and the QA model for MetaQA3, using just the QA data. In this experiment, we also varied $k$, the top-$k$ entities retrieved at each step. Figure 2 (left) shows pre-training the KB embeddings (fix-kge) consistently outperforms jointly training KB embeddings for the QA model (joint), and demonstrates that pre-training QE on simple KB queries can be useful for downstream tasks.

## 5   Concluding Discussion

EmQL is a new query embedding (QE) method, which combines a novel centroid-sketch representation for entity sets with neural retrieval over embedded KB triples. In this paper we showed that EmQL generalizes well, is differentiable, compact, scalable, and faithful with respect to deductive reasoning. However, there are areas for improvement. Compared to the reified KB method [3], EmQL learns and generalizes better, and does not rely on expensive sparse-matrix computations; however, unlike a reified KB, it requires KB-specific pretraining to find entity embeddings suitable for reasoning. Like most previous KBE or QE methods, EmQL sets must correspond to neighborhoods in embedding space[15]; EmQL's centroid-sketch representation additionally assumes that sets are of moderate cardinality. Finally, in common with prior QE methods [10, 17], EmQL does not support general logical negation, and has only very limited support for set difference.

In spite of these limitations, we showed that EmQL substantially outperforms previous QE methods in the usual experimental settings, and massively outperforms them with respect to faithfulness. In addition to improving on the best-performing prior QE method, we demonstrated that it is possible to incorporate EmQL as a neural module in a KBQA system: to our knowledge this is the first time that a QE system has been used to solve an downstream task (i.e., a task other than KB completion). Replacing a faithful localist representation of a KB with EmQL (but leaving rest of the QA system intact) leads to double-digit improvements in Hits@1 on three benchmark tasks, and leads to a new state-of-the-art on the two more difficult tasks.

## Broader Impact

**Overview.** This work addresses a general scientific question, query embedding (QE) for knowledge bases, and evaluates a new method, especially on a KB question-answering (KBQA) task. A key notion in the work the *faithfulness* of QE methods, that is, their agreement with deductive inference when the relevant premises are explicitly available. The main technical contribution of the paper is to show that massive improvements in faithfulness are possible, and that faithful QE systems can lead to substantial improvements in KBQA. In the following, we discuss how these advances may affect risks and benefits of knowledge representation and question answering technology.

**Query embedding.** QE, and more generally KBE, is a way of generalizing the contents of a KB by building a probabilistic model of the statements in, or entailed by, a KB. This probabilistic model finds statements that could plausibly true, but are not explicitly stored: in essence it is a noisy classifier for possible facts. Two risks need to be considered in any deployment of such technology: first, the underlying KB may contain (mis)information that would improperly affect decisions; second, learned generalizations may be wrong or biased in a variety of ways that would lead to improperly justified decisions. In particular, training data might reflect societal biases that will be thereby incorporated into model predictions. Uses of these technologies should provide audit trails and recourse so that their predictions can be explained to and critiqued by affected parties.

**KB question-answering.** General improvements to KBQA do not have a specific ethical burden, but like any other such technologies, their uses need to be subject to specific scrutiny. The general technology does require particular attention to accuracy-related risks. In particular, we propose a substantial "softening" of the typical KBQA architecture (which generally parses a question to produce a single hard KB query, rather than a soft mixture of embedded queries). In doing this we have replaced traditional KB, a mature and well-understood technology, with QE, a new and less well-understood technology. Although our approach makes learning end-to-end from denotations more convenient, and helps us reach a new state-of-the-art on some benchmarks, it is possible that replacing a hard queries to a KB with soft queries could lead to confusion as to whether answers arise from highly reliable KB facts, reliable reasoning over these facts, or are noise introduced by the soft QE system. As in KBE/QE, this has consequences for downstream tasks is uncertain predictions are misinterpreted by users.

**Faithful QE.** By introducing the notion of faithfullness in studies of approximate knowledge representation in QE, we provided a conceptual yardstick for examining the accuracy and predictability of such systems. In particular, the centroid-sketch formalism we advocate often allows one to approximately distinguish entailed answers vs generalization-based answers by checking sketch membership. In addition to quantitatively improving faithfulness, EmQL's set representation thus may qualitatively improve the interpretability of answers. We leave further validation of this conjecture to future work.

## Footnotes

[3]E.g., translational embedding schemes like TransE [1] would estimate the embedding for $y$ as $\hat{\mathbf{e}}_y = \mathbf{e}_x + \mathbf{e}_r$, where $\mathbf{e}_x$, and $\mathbf{e}_r$ are vectors embedding entity $x$ and relation $r$ respectively. Several other methods [9, 12] estimate $\hat{\mathbf{e}}_y = \mathbf{e}_x \mathbf{M}_r$ where $\mathbf{M}_r$ is a matrix representing $r$.

[4]If $\mathbf{v}_A, \mathbf{v}_B$ are $k$-hot vectors for sets $A, B$, then $\mathbf{v}_A + \mathbf{v}_B$ encodes $A \cup B$ and $\mathbf{v}_A \odot \mathbf{v}_B$ encodes $A \cap B$.

[5] While $\mathbf{a}_X$ could be based on other geometric representations for sets, we use MIPS queries because obtaining candidates this way can be very efficient [15].

[6] For count-min sketches, if $\mathbf{b}_\mathbb{I}$ is an all-ones matrix of the correct size, then $\forall i \ CM(i, \mathbf{b}_\mathbb{I}) = 1$.

[7]Of course, directly storing 100 embeddings is less useful for modeling, since that representation does not support operations like relation following or intersection.

[8]The sketch is not used for this objective, but is used in § 4.1 where we train a QA system which includes EmQL as a component. Hence in general it is necessary for inference with the sketch to be differentiable.

[9] Of ccourse, test queries are always distinct from training queries.

[10] In particular, intermediate EmQL representations are never "decoded" by converting them to entity lists.

[11] Here $d = 2000$ was the largest value of $d$ supported by our GPUs. Note this is still much smaller than the number of entities, which would be number required to guarantee arbitrary sets could be memorized with boxes.

[12]Here $A - B$ is a non-compositional *set difference* operator, see Supplementary Materials for details.

[13]An important difference is that in ReifKB $R_1$ and $R_2$ are $k$-hot representations of sets of relation ids, not centroids in embedding space.

[14]The relational filtering operation is not defined for ReifKB, although it could be implemented with sequences of simpler operations.

[15]A notable exception is Query2Box which represents sets with unions of rectangles using a non-compositional set union operator. Although non-compositional set union could be added to EmQL it is currently not implemented.

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
