[Supplementary Material]

# Faithful Embeddings for Knowledge Base Queries: Supplementary Material

**Haitian Sun**    **Andrew O. Arnold**[*]    **Tania Bedrax-Weiss**
**Fernando Pereira**    **William W. Cohen**
Google Research
{haitiansun,tbedrax,pereira,wcohen}@google.com
AWS AI
anarnld@amazon.com

## A  Notation

The notation used in this paper is summarized in Table 1.

| | |
|---|---|
| $W, X, Y$ | sets of entities |
| $R$ | set of relations |
| $r$ | a single relation |
| $x, y$ | entities |
| $x_i$ | entity with index $i$ |
| | |
| $A, B$ | set of anything (entities or relations) |
| $U$ | universal set |
| $\mathbf{v}_A$ | a $k$-hot vector for a set $A$ |
| | |
| $r(x, y)$ | asserts this triple is in the KB |
| $\mathbf{E}$ | matrix of entity embeddings |
| $\mathbf{e}_x, \mathbf{e}_r$ | embedding of entity $x$, relation $r$ |
| $\mathbf{e}_i$ | embedding of entity with index $i$, i.e. $\mathbf{e}_i = \mathbf{E}[i,:]$ |
| **KB** | matrix of triple embeddings, i.e., row for $r(x, y)$ is $[\mathbf{e}_r; \mathbf{e}_x; \mathbf{e}_y]$ |
| | |
| $(\mathbf{a}_X, \mathbf{b}_X)$ | area and sketch that represent set $X$ |
| $CM(i, \mathbf{b})$ | score for entity $i$ in the count-min sketch $\mathbf{b}$ |
| $X.follow(R)$ | soft version of $\{y \mid \exists r \in R, x \in X : r(x, y)\}$ |
| $X.filter(R, Y)$ | soft version of $\{x \in X \mid \exists r \in R, y \in Y : r(x, y)\}$ |

Table 1: Notation used in the paper, excluding notation used only in § B

## B  Background on count-min sketches

### B.1  Definitions

Count-min sketches [1] are a widely used randomized data structure. We include this discussion for completeness, and our analysis largely follows [2].

A count-min sketch, as used here, is an approximation of a vector representation of a weighted set. Assume a universe $U$ which is a set of integer "object ids" from $\{1, \ldots, N\}$. A set $A \subseteq U$ can be

---

[*]Work done while at Google Research.

encoded as a vector $\mathbf{v}_A \in \mathbb{R}^n$ such that $\mathbf{v}_A[i] = 0$ if $i \notin S$, and otherwise $\mathbf{v}_A[i]$ is a real-numbered weight for entity $i$ in set $S$. The purpose of the count-min sketch is to approximate $\mathbf{v}_A$ with limited storage.

Let $h$ be a hash function mapping $\{1, \ldots, N\}$ to a smaller range of integers $\{1, \ldots, N_W\}$, where $N_W \ll N$. The *primitive sketch of* $\mathbf{v}_A$ *under* $h$, written $\mathbf{s}_h(\mathbf{v}_A)$, is a vector such that

$$\mathbf{s}_h(\mathbf{v}_A)[j] = \sum_{i:h(i)=j} \mathbf{v}_A[i]$$

Algorithmically, this vector could be formed by starting with an all-zero's vector of length $N_W$, then looping over every pair $(i, w_i)$ where $w_i = \mathbf{v}_A[i]$ and incrementing each $\mathbf{s}_h[j]$ by $w_i$. A primitive sketch $\mathbf{s}_h$ contains some information about $\mathbf{v}_A$: to look up the value $\mathbf{v}_A[i]$, we could look up $\mathbf{s}_h[h(i)]$, and this will have the correct value if no other set element $i'$ hashed to the same location. We can improve this by using multiple hash functions.

Specifically, let $H = \{h_1, \ldots, h_{N_D}\}$ be a list of $N_D$ hash functions mapping $\{1, \ldots, N\}$ to the smaller range of integers $\{1, \ldots, N_W\}$. The *count-min sketch* $\mathbf{S}_H(\mathbf{v}_A)$ *for a* $\mathbf{v}_A$ *under* $H$ is a matrix such that each row $j$ is the primitive sketch of $\mathbf{v}_A$ under $h_j$. This sketch is an $N_W \times N_D$ matrix: $N_W$ is called the sketch width and $N_D$ is called the sketch depth.

Let $\mathbf{S}$ be the count-min sketch for $A$. To "look up" (approximately recover) the value of $\mathbf{v}_A[i]$, we compute this quantity

$$CM(i, \mathbf{S}) \equiv \min_{j=1}^{N_D} \mathbf{S}[j, h_j(i)]$$

In other words, we look up the hashed value associated with $i$ in each of the $N_D$ primitive sketches, and take the minimum value.

## B.2   Linearity and implementation nodes

Count-min sketches also have a useful "linearity" property, inherited from primitive sketches. It is easy to show that for any two sets $A$ and $B$ represented by vectors $\mathbf{v}_A$ and $\mathbf{v}_B$

$$\begin{aligned}
\mathbf{S}_H(\mathbf{v}_A + \mathbf{v}_B) &= \mathbf{S}_H(\mathbf{v}_A) + \mathbf{S}_H(\mathbf{v}_B) \\
\mathbf{S}_H(\mathbf{v}_A \odot \mathbf{v}_B) &= \mathbf{S}_H(\mathbf{v}_A) \odot \mathbf{S}_H(\mathbf{v}_B)
\end{aligned}$$

Here, as elsewhere in this paper, $\odot$ is Hadamard product.

In general, although it is mathematically convenient to define the behavior of sketches in reference to $k$-hot vectors, it is *not necessary to construct a vector* $\mathbf{v}_A$ *to construct a sketch*: all that is needed is the non-zero weights of the elements of $A$. Alternatively, if one precomputes and stores the sketch for each singleton set, it is possible to create sketches for an arbitrary set by gathering and sum-pooling the sketches for each element.

## B.3   Probabilistic bounds on accuracy

We assume the hash functions are random mappings from $\{1, \ldots, N\}$ to $\{1, \ldots, N_W\}$. More precisely, we assume that for all $i \in \{1, \ldots, N\}$, and all $j \in \{1, \ldots, N_W\}$, $\Pr(h_i(x) = a) = \frac{1}{N_W}$. We will also assume that the $N_D$ hash functions are are all drawn *independently* at random. More precisely, for all $i \neq i'$, $i, i' \in \{1, \ldots, N\}$, all $j, j' \in \{1, \ldots, N_D\}$ and all $k, k' \in \{1, \ldots, N_W\}$, $\Pr(h_j(i) = k \wedge h_{j'}(i') = k') = \frac{1}{N_W^2}$.

Under this assumption, the probability of errors can be easily bounded. Suppose the sketch width is at least twice the cardinality of $A$, i.e., $|A| < m$ and $N_W > 2m$. Then one can show for all primitive sketches $j$:

$$\Pr(\mathbf{S}[j, h_j(i)] \neq \mathbf{v}_A[i]) \leq \frac{1}{2}$$

From this one can show that the probability of any error in a count-min sketch decreases exponentially in sketch depth. (This result is a slight variant of one in [2].)

**Theorem 1** *Assuming hash functions are random and independent as defined above, then if **S** is a count-min sketch for A of depth $N_D$, and $N_W > 2|A|$, then*

$$\Pr(CM(\boldsymbol{S}, i) \neq \boldsymbol{v}_A[i]) \; \leq \; \frac{1}{2^{N_D}}$$

This bound applies to a single CM operation. However, by using a union bound it is easy to assess the probability of making an error in any of a series of CM operations. In particular, we consider the case that there is some set of candidates $C$ including all entities in $A$, i.e., $A \subseteq C \subseteq U$, and consider recovering the set $A$ by performing a CM lookup for every $i' \in C$. Specifically, we say that $A$ *can be recovered from **S** using C* if $A \subseteq C$ and

$$\forall i' \in C, CM(i', \mathbf{S}) = \mathbf{v}_A[i']$$

Note that this implies the sketch must correctly score every $i' \in C - A$ as zero. Applying the union bound to Theorem 1 leads to this result.

**Theorem 2** *Let **S** be a count-min sketch for A of depth $N_D$ and with $N_W > 2|A|$, and let $C \supseteq A$. If $N_D > \log_2 \frac{|C|}{\delta}$ then with probability at least 1-δ, A can be recovered from **S** using C.*

Many other bounds are known for count-min sketches: perhaps the best-known result is that for $N_W > \frac{2}{\epsilon}$ and $N_D > \log \frac{1}{\delta}$, the probability that $CM(i, \mathbf{S}) > \mathbf{v}_A[i] + \epsilon$ is no more than $\delta$ [1]. Because there are many reasonable formal bounds that might or might not apply in an experimental setting, typically the sketch shape is treated as a hyperparameter to be optimized in experimental settings.

## C  Set difference

Another operation we use is set difference: e.g. "movie directors but not writers" requires one to compute a set difference $A_{\text{directors}} - B_{\text{writers}}$. In computing a set difference, the soft-type of the output $A - B$ is the same as that of $A$, and we exclude the necessary elements from the count-min sketch to produce $(\mathbf{a}_{A-B}, \mathbf{b}_{A-B})$, where

$$\mathbf{a}_{A-B} = \mathbf{a}_A$$
$$\mathbf{b}_{A-B} = \mathbf{b}_A \odot (\mathbf{b} \neq 0)$$

This is exact when $B$ is unweighted (the case we consider here), but only approximates set difference for general weighted sets.

## D  More experiment details

### D.1  Learn to reason over a KB

The statistics of the Query2Box datasets are shown in Table 2.

We also measure the MRR on the Query2Box datasets. The results are presented in Table 3 and 4.

### D.2  Question answering

#### D.2.1  Datasets

The statistics of MetaQA and WebQuestionsSP datasets are listed in Table 5. For WebQuestionsSP, we used a subset of Freebase obtained by gathering triples that are within 2-hops of the topic entities in Freebase. We exclude a few extremely common entities and restrict our KB subset so there are at most 100 tail entities for each subject/relation pair (reflecting the limitation of our model to sets of cardinality less than 100).

#### D.2.2  MetaQA

MetaQA makes use of the set difference operation. For example, to answer the question "What are other movies that have the same director as *Inception*?", we need to first find the director of *Inception*, *Christopher Nolan*, and all movies directed by him. Since the question above asks about

|  | Entities | Relations | Training Triples | Test Triples | Total Triples |
|---|---|---|---|---|---|
| FB15k | 14,951 | 1,345 | 533,142 | 59,071 | 592,213 |
| FB15k-237 | 14,505 | 237 | 289,641 | 20,438 | 310,079 |
| NELL995 | 63,361 | 200 | 128,537 | 14,267 | 142,804 |

(a) Size of splits into train and test for all the Query2Box KBs.

|  | Train | | | Test | |
|---|---|---|---|---|---|
| task | Basic sets | Follow (1p) | Intersection | Follow (1p) | Others |
| FB15k | 11,611 | 96,750 | 355,966 | 67,016 | 8,000 |
| FB15k-237 | 11,243 | 50,711 | 191,934 | 22,812 | 5,000 |
| NELL995 | 19,112 | 36,469 | 108,958 | 17,034 | 4,000 |

(b) Number of training and testing examples of the Query2Box datasets. Training data for EmQL are derived from the same training KB as Query2Box. EmQL is directly evaluated on the same test data without further fine-tuning.

Table 2: Statistics for the Query2Box datasets.

| *generalization* | | 1p | 2p | 3p | 2i | 3i | ip | pi | 2u | up | Avg |
|---|---|---|---|---|---|---|---|---|---|---|---|
| FB15k | GQE | 63.6 | 34.6 | 25.0 | 51.5 | 62.4 | 15.1 | 31.0 | 37.6 | 27.3 | 38.7 |
|  | Q2B | **78.6** | 41.3 | 30.3 | 59.3 | 71.2 | 21.1 | 39.7 | **60.8** | 33.0 | 48.4 |
|  | +*d*=2000 | 54.3 | 32.0 | 27.0 | 35.5 | 50.7 | 13.7 | 27.0 | 44.1 | 26.3 | 34.5 |
|  | EmQL(ours) | 42.4 | **50.2** | **45.9** | **63.7** | **70.0** | **60.7** | **61.4** | 9.0 | **42.6** | **49.5** |
|  | - sketch | 50.6 | 46.7 | 41.6 | 61.8 | 67.3 | 54.2 | 53.5 | 21.6 | 40.0 | 48.6 |
| FB15k-237 | GQE | 40.5 | 21.3 | 15.5 | 29.8 | 41.1 | 8.5 | 18.2 | 16.9 | 16.3 | 23.1 |
|  | Q2B | **46.7** | 24 | 18.6 | 32.4 | 45.3 | 10.8 | 20.5 | **23.9** | 19.3 | 26.8 |
|  | +*d*=2000 | 37.2 | 20.7 | 19.4 | 22.6 | 37.1 | 9.7 | 16.8 | 20.0 | 17.8 | 22.4 |
|  | EmQL(ours) | 37.7 | **34.9** | **34.3** | **44.3** | **49.4** | **40.8** | **42.3** | 8.7 | 28.2 | **35.8** |
|  | - sketch | 43.1 | 34.6 | 33.7 | 41.0 | 45.5 | 36.7 | 37.2 | 15.3 | **32.5** | 35.5 |
| NELL995 | GQE | 41.8 | 23.1 | 20.5 | 31.8 | 45.4 | 8.1 | 18.8 | 20.0 | 13.9 | 24.8 |
|  | Q2B | **55.5** | 26.6 | 23.3 | 34.3 | 48.0 | 13.2 | 21.2 | **36.9** | 16.3 | 30.6 |
|  | +*d*=2000 | 49.1 | 22.1 | 17.5 | 21.4 | 39.9 | 8.9 | 17.2 | 26.4 | 8.1 | 23.4 |
|  | EmQL(ours) | 41.5 | **40.4** | **38.6** | **62.9** | **74.5** | **49.8** | **64.8** | 12.6 | 35.8 | **46.8** |
|  | - sketch | 48.3 | 39.5 | 35.2 | 57.2 | 69.0 | 48.0 | 59.9 | 25.9 | **38.2** | **46.8** |
| *entailment* | | | | | | | | | | | |
| FB15k | Q2B | 68.0 | 39.4 | 32.7 | 48.5 | 65.3 | 16.2 | 32.9 | 61.4 | 28.9 | 43.7 |
|  | +*d*=2000 | 59.0 | 36.8 | 30.2 | 40.4 | 57.1 | 14.8 | 28.9 | 49.2 | 28.7 | 38.3 |
|  | EmQL(ours) | **98.5** | **96.3** | **91.1** | **91.4** | **88.1** | **87.8** | **89.2** | **88.7** | **91.3** | **91.4** |
|  | - sketch | 85.1 | 50.8 | 42.4 | 64.4 | 66.1 | 50.4 | 53.8 | 43.2 | 42.7 | 55.5 |
| FB15k-237 | Q2B | 58.5 | 34.3 | 28.1 | 44.7 | 62.1 | 11.7 | 23.9 | 40.5 | 22.0 | 36.2 |
|  | +*d*=2000 | 50.7 | 30.1 | 26.1 | 34.8 | 55.2 | 11.4 | 20.6 | 32.8 | 21.5 | 31.5 |
|  | EmQL(ours) | **100.0** | **99.5** | **94.7** | **92.2** | **88.8** | **91.5** | **93.0** | **94.7** | **93.7** | **94.2** |
|  | - sketch | 89.3 | 55.7 | 39.9 | 62.9 | 63.9 | 51.9 | 54.7 | 53.8 | 44.7 | 57.4 |
| NELL995 | Q2B | 83.9 | 57.7 | 47.8 | 49.9 | 66.3 | 19.9 | 29.6 | 73.7 | 31.0 | 51.1 |
|  | +*d*=2000 | 75.7 | 49.9 | 36.9 | 40.5 | 60.1 | 17.1 | 25.6 | 63.5 | 24.4 | 43.7 |
|  | EmQL(ours) | **99.0** | **99.0** | **97.1** | **99.7** | **99.6** | **98.7** | **98.9** | **98.8** | **98.5** | **98.8** |
|  | - sketch | 94.5 | 77.4 | 52.9 | 97.4 | 97.5 | 88.1 | 90.8 | 70.4 | 73.5 | 82.5 |

Table 3: Detailed Hits@3 results for all the Query2Box datasets.

*other* movies, the model should also remove the movie *Inception* from this set to obtain the final answer set $Y$. Thus in the first line of our model, we write

$$\hat{Y} = X_q.follow(R_1).follow(R_2) - X_q$$

For MetaQA, the entity embedding is just a learned lookup table. The question representation $encode(q)$ is computed with a bag-of-word approach, i.e., an average pooling on the word embeddings of question $q$. The embedding size is 64, and scaling parameter for relation $\lambda$ is 1.0. Our count-min sketch has depth $N_D = 20$ and width $N_W = 500$. We set $k = 100$ to be the number of entities we retrieve at each step, and we pre-train KB embeddings and fix the embeddings when training our QA model.

| *generalization* | | 1p | 2p | 3p | 2i | 3i | ip | pi | 2u | up | Avg |
|---|---|---|---|---|---|---|---|---|---|---|---|
| FB15k | GQE | 0.505 | 0.320 | 0.222 | 0.439 | 0.536 | 0.142 | 0.280 | 0.300 | 0.242 | 0.332 |
| | Q2B | **0.654** | 0.373 | 0.274 | 0.488 | 0.602 | 0.194 | 0.339 | **0.468** | 0.301 | 0.410 |
| | +$d$=2000 | 0.461 | 0.289 | 0.242 | 0.292 | 0.421 | 0.130 | 0.236 | 0.342 | 0.235 | 0.294 |
| | EmQL(ours) | 0.368 | **0.452** | **0.409** | **0.574** | **0.609** | **0.556** | **0.538** | 0.074 | **0.375** | **0.439** |
| | - sketch | 0.453 | 0.418 | 0.362 | 0.556 | 0.592 | 0.503 | 0.482 | 0.182 | 0.351 | 0.433 |
| FB15k-237 | GQE | 0.346 | 0.193 | 0.145 | 0.250 | 0.355 | 0.086 | 0.156 | 0.145 | 0.151 | 0.203 |
| | Q2B | **0.400** | 0.225 | 0.173 | 0.275 | 0.378 | 0.105 | 0.18 | **0.198** | 0.178 | 0.235 |
| | +$d$=2000 | 0.322 | 0.196 | 0.185 | 0.193 | 0.318 | 0.095 | 0.149 | 0.174 | 0.166 | 0.200 |
| | EmQL(ours) | 0.334 | **0.305** | **0.304** | **0.378** | **0.436** | **0.351** | **0.358** | 0.075 | 0.241 | **0.309** |
| | - sketch | 0.370 | 0.297 | 0.306 | 0.345 | 0.400 | 0.311 | 0.306 | 0.129 | **0.272** | 0.304 |
| NELL995 | GQE | 0.311 | 0.193 | 0.175 | 0.275 | 0.408 | 0.080 | 0.170 | 0.159 | 0.130 | 0.211 |
| | Q2B | **0.413** | 0.227 | 0.208 | 0.288 | 0.414 | 0.125 | 0.193 | **0.266** | 0.155 | 0.254 |
| | +$d$=2000 | 0.308 | 0.174 | 0.151 | 0.171 | 0.350 | 0.083 | 0.150 | 0.183 | 0.087 | 0.184 |
| | EmQL(ours) | 0.372 | **0.351** | **0.349** | **0.539** | **0.654** | **0.441** | **0.561** | 0.105 | 0.311 | **0.409** |
| | - sketch | 0.431 | 0.349 | 0.300 | 0.493 | 0.588 | 0.423 | 0.527 | 0.22 | **0.324** | 0.406 |
| *entailment* | | | | | | | | | | | |
| FB15k | Q2B | 0.559 | 0.347 | 0.288 | 0.389 | 0.553 | 0.145 | 0.280 | 0.444 | 0.257 | 0.362 |
| | +$d$=2000 | 0.498 | 0.327 | 0.274 | 0.336 | 0.492 | 0.139 | 0.251 | 0.386 | 0.257 | 0.329 |
| | EmQL(ours) | **0.983** | **0.961** | **0.908** | **0.908** | **0.872** | **0.881** | **0.883** | **0.887** | **0.910** | **0.910** |
| | - sketch | 0.819 | 0.448 | 0.368 | 0.564 | 0.580 | 0.420 | 0.466 | 0.385 | 0.383 | 0.492 |
| FB15k-237 | Q2B | 0.476 | 0.301 | 0.249 | 0.364 | 0.638 | 0.113 | 0.207 | 0.311 | 0.203 | 0.318 |
| | +$d$=2000 | 0.432 | 0.262 | 0.233 | 0.292 | 0.466 | 0.109 | 0.183 | 0.255 | 0.198 | 0.270 |
| | EmQL(ours) | **0.998** | **0.988** | **0.949** | **0.902** | **0.867** | **0.892** | **0.909** | **0.947** | **0.934** | **0.932** |
| | - sketch | 0.861 | 0.504 | 0.352 | 0.554 | 0.581 | 0.451 | 0.475 | 0.499 | 0.400 | 0.520 |
| NELL995 | Q2B | 0.652 | 0.465 | 0.412 | 0.420 | 0.562 | 0.186 | 0.257 | 0.516 | 0.269 | 0.415 |
| | +$d$=2000 | 0.545 | 0.409 | 0.331 | 0.357 | 0.526 | 0.155 | 0.217 | 0.399 | 0.253 | 0.355 |
| | EmQL(ours) | **0.990** | **0.990** | **0.971** | **0.996** | **0.996** | **0.987** | **0.987** | **0.988** | **0.985** | **0.988** |
| | - sketch | 0.939 | 0.750 | 0.462 | 0.952 | 0.954 | 0.851 | 0.871 | 0.653 | 0.702 | 0.793 |

Table 4: MRR results on the Query2Box datasets.

| | Train | Dev | Test |
|---|---|---|---|
| MetaQA 2-hop | 118,980 | 14,872 | 14,872 |
| MetaQA 3-hop | 114,196 | 14,274 | 14,274 |
| WebQuestionsSP | 2,848 | 250 | 1,639 |

(a) Number of train/dev/test data

| | Triples | Entities | Relations |
|---|---|---|---|
| MetaQA | 392,906 | 43,230 | 18 |
| WebQuestionsSP | 1,352,735 | 904,938 | 695 |

(b) Size of KB

Table 5: Statistics for the MetaQA and WebQuestionsSP datasets.

### D.2.3   WebQuestionsSP

We use pre-trained BERT to encode our question $q$, i.e., $encode(q)$ is the BERT embedding of the [CLS] token. The relation sets $R_1$, $R_2$, $R_3$ are linear projections of the question embedding $encode(q)$ paired with a vacuous all-ones sketch $\mathbf{b}_\mathbb{I}$. Relation centroids are stacked with one extra dimension that encodes the hard-type of entities: here the hard-type is a binary value that indicates if the entity is a $cvt$ node or not.

For this dataset, to make the entities and relations easier to predict from language, the embedding of each entity was adapted to include a transformation of the BERT encoding of the surface form of the entity names. Let $\mathbf{e}_x^0$ be the embedding of the [CLS] token from a BERT [3] encoding of the canonical name for entity $x$, and let $\mathbf{e}_x^1$ be a vector unique to $x$. Our pre-trained embedding for $x$ is then $\mathbf{e}_x = \left[W^T\mathbf{e}_x^0; \mathbf{e}_x^1\right] p$, where $W$ is a learned projection matrix. The embedding of relation $r$ is set to the BERT encoding ([CLS] token) of the canonical name of relation $r$. In this experiments the BERT embeddings are transformed to 128 dimensions and the entity-specific portion $\mathbf{e}_x^1$ has a dimension of 32. The scaling parameter for relation $\lambda$ is 0.1.

The KB embedding is fixed after pre-training. We use a count-min sketch with depth $N_D = 20$ and width $N_W = 2000$, and we retrieve $k = 1000$ intermediate results at each step.

In the ablation study, we did two more experiments on the WebQuestionsSP dataset. First, we remove the BERT pre-trained embedding, and instead randomly initialize the KB entity and relation embeddings, and train the set operations. The performance of EmQL (no-bert) on the downstream QA task is 1.3% lower than our full model. Second, we replace the exact MIPS with a fast maximal inner-product search [4]. This fast MIPS is an approximation of MIPS that eventually causes 2.1% drop in performance (Table 6).

|                     | WebQuestionsSP |
|---------------------|:--------------:|
| EmQL                | **75.5**       |
| EmQL (no-sketch)    | 53.2           |
| EmQL (no-filter)    | 65.2           |
| EmQL (approx. MIPS) | 73.4           |
| EmQL (no-bert)      | 74.2           |

Table 6: Ablation study on WebQuestionsSP