[Reviews · NeurIPS 2020]

Review 1

Summary and Contributions: The authors propose a novel QE method to solve the problem that existing QE systems may disagree with deductive reasoning on answers that do not require generalization or relaxation. The proposed QE method improves the faithfulness of the deductive reasoning significantly, besides, the neural question-answering which plugging the new QE module outperforms the state-of-the-art system substantially.

Strengths: 1. This article defines a reasoning system framework, which can be seamlessly integrated with the end-to-end model. The work is very innovative. The theoretical brief is quite sufficient, the definitions of related symbols and formulas are relatively complete, and the claim is relatively powerful. 2. In addition, this work is also very reasonable in terms of experimental design. First, the authors verify the ability of generalization and entailment of the proposed QE method through learning to reason with a KB. Secondly, the proposed QE module is plugged into the Question Answering system, the experimental results show that, on MetaQA3 and WebQSP datasets the proposed QE module exceed the state-of-the-art by a large margin. 3. This work also has a relatively large impact on the research work of the NeurIPS community in this area.

Weaknesses: 1. The disadvantage of EmQL’s approach is that unions are only allowed between sets of similar “soft types”. In fact, EmQL’s centroid-sketch representation will not compactly encode any set of sufficiently diverse entities. 2. The authors mentioned that the limitation of EmQL could be addressed by introducing a second normal-form disjunction zoperation that outputs a union of centroid-sketch representations, much as Query2Box’s disjunction 173 outputs a union of boxes. However, we leave such an extension as a topic for future work. 3. For the results in the Table 4, the authors claim that the results on MetaQA2 is comparable to the previous state-of-the-art. However, the result (98.6) is lower than the state-of-the-art (99.9 produced by the PullNet), I think the authors should give more detailed explanation about the reason.

Correctness: The author’s conclusion in the paper seems to be reasonable, and the empirical methodology is also correct in the general direction. The experimental design is reasonable, and the experimental results are very convincing to draw conclusions.

Clarity: The writing of this paper is good and native, however, there still are some typos.

Relation to Prior Work: The authors mention enough related works in the paper. The most important previous work is related to the Query2Box method. The proposed method is very different from the previous methods. The author also explained in detail the differences from the Query2Box method. In addition, the authors also introduce related work in detail, and also elaborated and compared the advantages and disadvantages of the previous work.

Reproducibility: Yes

Additional Feedback: 1. Below the line 149, if the K contains a row r_t = [e_r; e_x; e_y], the K may be a very huge matric, because the combination of the e_r, e_x and e_y may be very large, right? 2. In Table 4, why the results on MetaQA2 datasets is worse than the previous state-of-the-art produced by the PullNet method? This is not consistent with those on the other two datasets. 3. Line 292, fix-kge should be fix-kbe, right? 4. Figure 2 shows that pre-training the KB embeddings (fix-kge) consistently outperforms jointly training KB embeddings for the QA model. That will be better if some explanations are given. 5. I hope that some case studies are given to show why the pre-training the KB embeddings could improve the performance of the Question answering system.


Review 2

Summary and Contributions: This paper presents an embedding based neural query language, EmQL, that generalizes to the unknown facts in a KB and also performs logical entailments better than existing methods like Qeury2Box. A neural way of executing logical entailment queries is not novel, e.g. Cohen et al in [3] introduced NQL. EmQL builds on top of NQL by embedding entities and relations in low dimensional space and uses count min sketches to represent a set of entities/relations in a novel way. It obtains better generalization while also retaining NQL’s performance on logical entailment queries. Contributions: 1. The work borrows ideas from NQL and obtains both generalization and faithfulness to logical entailment queries. Demonstrated by achieving better performance than other query embedding methods like Query2Box and Graph Query Embeddings (GQE). 2. A novel way of representing a set of entities in an embedding space via an average entity embedding and a count-min sketch to identify the actual entities in the set that are close to average embedding. 3. First work to train a neural KBQA model end to end, exploiting the fully differentiable nature of EmQL and achieves sota results ===================== I acknowledge the rebuttal. My rating remains unchanged.

Strengths: 1. End to end differentiable reasoning: Typical neuro symbolic systems use non-differentiable reasoning methods, making it difficult to train the neural model, and usually, RL based techniques have to be used for training. NQL made the reasoning task differentiable and this work, EmQL, takes it a step further by achieving better generalization. 2. Faithfulness and generalization: Fares better than other query embeddings methods on the tradeoff between faithfulness and generalization. 3. Empirical evaluation: achieves sota results in KBQA tasks.

Weaknesses: 1. The most crucial novelty of the paper is the use of count min sketches to represent set of entities, without which the set representation is nothing but bag-of-words embedding along with nearest neighbor search. Therefore, count min sketches should be covered in detail in the paper and not in the supplementary. 2. Q2Box can naturally represent sets of different sizes via boxes of different sizes, however EmQL first extracts top k elements (k is fixed), irrespective of the size of the set and further pruning is done based on count min sketch. 3. Empirical evaluation: A. Why not compare against Q2Box in KBQA experiments? B. It would have been better if there is some experiment to judge the efficacy of count-min sketches to represent sets, e.g. take a set of entities with same soft types, encode it using sparse dense representations, decode it, and then compare the original set with the decoded version. Such experiments would validate the use of count-min sketch to represent sets.

Correctness: Yes

Clarity: Yes

Relation to Prior Work: Yes

Reproducibility: Yes

Additional Feedback: 1. When vacuous sketches are used in the intermediate steps, e.g. in R1 in MetaQA model, what is the intermediate output? Is it the dense-sparse representation of the entities in top-k facts? Isn’t that a problem when k is large? 2. Both union and intersection of two sets have the same centroid and with a vacuous sketch, as in KBQA experiments, both collapse to the same representation. Won’t this be an issue in case there is a template that requires intersection as well in addition to unions? 3. For a given query, EmQL ranks all the entities (or gives a distribution over entities) instead of explicitly giving a set as an answer. This becomes an issue when the ground truth is a set of different cardinality for different queries. Returning top k for all queries doesn’t make sense, especially when k is deliberately chosen to be large to achieve high recall. How can we modify EmQL for such scenarios? 4. The workings of count min sketch haven’t been discussed in the paper in detail. Would prefer if some discussion from the appendix moves to the main paper. 5. Finally, EmQL can be used for the task of KBC as well. It would be interesting to see how it fares against sota methods for KBC. 6. Typos: a. Line #66: X.follow(r) should be X.follow(R) b. Line #139: specifically “it” can be .. c. Line #143: spelling of Hadamard. D. Above Line #153: W and X are used interchangeably, the LHS of equation should have “X.filter(R,Y)” instead of “W.filter(R,Y)”


Review 3

Summary and Contributions: The paper introduces EmQL, a way to make soft queries over a neural KB which is substantially more faithful (self-consistent) than earlier approaches. They show significant improvements on certain downstream tasks as well.

Strengths: This seems like a novel approach to bringing inductive bias from exact KB intersection and entailment operations into a differentiable setting, using approximate compressed representations of weighted entity sets. The approach is described quite clearly, even for a non-expert like myself, with useful background material in the supplementary material. The experiments show substantial improvements over the referenced existing methods, especially when it comes to inferences over existing KB triples (as opposed to the ones inferred as part of KB completion). There are also seemingly notable gains on the downstream tasks of MetaQA and WebQuestionsSP.

Weaknesses: As described in the paper, there is still an underlying assumption that the relevant sets of entities are well described by regions around their centroids in embedding space.

Correctness: To the best of my judgement, as a non-expert, the claims and methods seem correct.

Clarity: The paper is quite clear, with helpful background and details in the supplementary material. Maybe some more explicit examples from the downstream tasks could help clarify qualitatively what improvements have been obtained.

Relation to Prior Work: There are relevant comparisons to earlier work.

Reproducibility: Yes

Additional Feedback: =========== After author feedback: Thank you for the feedback, the suggested changes sound good. My score is unchanged.

[Author Response · NeurIPS 2020]

We thank the reviewers for their helpful comments. We will move the count-min sketch details to the main paper and add more discussion on experiment results. We will fix the typos pointed out by the reviewers.

To Reviewer 1 and 3:

- *"... EmQL's centroid-sketch representation will not compactly encode any set of sufficiently diverse entities."* and *"assumption that the relevant sets of entities are well described by regions around their centroids in embedding space ..."* In EmQL representation, we assume that elements in the set are similar to each other. While it restricts the power of representing arbitrary sets, this assumption holds in many compositional KB reasoning tasks. With proper pretraining, it enables to accurately retrieve a small subset of candidates and efficiently check their membership with a count-min sketch. Efficiently representing arbitrary sets is a very interesting and challenging task. We would like to leave it for future exploration.

To Reviewer 1:

- *"... if the K contains a row $r_t = [e_r; e_x; e_y]$, the K may be a very huge matric ..."* In this paper, fact embeddings are encoded as a concatenation of its subject, relation, and object embeddings for simplicity. One can easily design more powerful encoding methods to get more compact fact embeddings to save memory. Even though the fact embedding table is large, EmQL is very scalable at inference time with the CPU-based fast Maximum Inner Product Search (MIPS) algorithms, e.g. Faiss (Johnson et al, 2017).
- *"... why the results on MetaQA2 datasets is worse than ... the PullNet method?"* PullNet is a complicated iterative "retrieve and classify" model for multi-hop reasoning trained with distantly supervised labels at intermediate steps. The retrieval on the next step is conditioned on the previous steps. EmQL treats each step independently for simplicity. Intermediate labels are not required to train EmQL.

To Reviewer 2:

- *"Why not compare against Q2Box in KBQA experiments?"* The KBQA problem is modeled as $Y = X.follow(R)$ where $R$ is a weighted set of relations. Query2Box embedding is designed to follow a specified relation $r$. It's not clear how to follow a set of relations $R$ with learned relation weights.
- *"... some experiment to judge the efficacy of count-min sketches to represent sets ..."* The entailment experiment results in Table 2 show the ablation study without sketch (EmQL - sketch). Without the count-min sketch, the model performs more than 30% worse in Hits@3 on complicated queries, and 10% worse on simple queries. We run the set decoding experiments proposed by the reviewer. On FB15k-237, we get 99.8% F1 on sets with less than 100 elements (with k = 1000).
- *"When vacuous sketches are used in the intermediate steps ... what is the intermediate output?"* Correct, the intermediate output is the dense-sparse representation of entities returned from the top k facts. When k is large, we should increase the width of the count-min sketch to accommodate more entities. In general, it should be larger than 2 * k to minimize the risk of collision. Increasing k helps improve the recall but can decrease the efficiency.
- *"Both union and intersection of two sets have the same centroid and with a vacuous sketch ..."* Vacuous sketches are often applied on relations in KBQA tasks when relations on the inferential chain are learned. For union (intersection) of two sets of entities, the sketches of the two sets are usually present from the previous steps. The sketch of the unioned (intersected) set is then derived from the two sketches. It is possible to get a vacuous sketch for the unioned (intersected) set in case that the sketches of both sets are vacuous. We would expect performance to drop in such cases.
- *"... however EmQL first extracts top k elements (k is fixed), irrespective of the size of the set ..."* and *"This becomes an issue when the ground truth is a set of different cardinality for different queries ..."* An alternative of top k retrieval is to apply soft thresholding to the ranking scores of all elements. This requires computing ranking scores for all entries in the KB and can be very inefficient when KB is large. We compromise with the top k operation that is fairly efficient with the CPU-based approximate Maximum Inner Product Search (MIPS) (Johnson et al, 2017).
- *"EmQL can be used for the task of KBC ..."* Yes, it's interesting to experiment with EmQL on KBC tasks. We would like to leave this for future work.

[1] J. Johnson, M. Douze, and H. Jégou. Billion-scale similarity search with GPUs. arXiv preprint arXiv:1702.08734, 2017

[Meta-Review · NeurIPS 2020]

This paper presents an embedding based neural query language called EmQL, that generalizes to the unknown facts in KB and also performs logical entailments better than existing methods like Qeury2Box. Strength • The proposed method is sound and novel. • Experimental results are convincing. • The paper is clearly written. Weakness • There are underlying assumption in the proposed approach. • The presentation can be further improved.